# CAN DATA-DRIVEN MACHINE LEARNING REACH SYMBOLIC-LEVEL LOGICAL REASONING? – THE LIMIT OF THE SCALING LAW

## ABSTRACT

With the qualitative extension of embedding representation and the method of explicit model construction, neural networks may achieve the rigour of symbolic level logic reasoning without training data, raising the question of where the limit of the scaling law for logical reasoning lies, i.e., whether data-driven machine learning systems can achieve the same level by increasing training data and training time. We show two methodological limitations that prevent supervised deep learning from reaching the symbolic-level syllogistic reasoning, a foundational subset of logical reasoning: (1) training data can not distinguish all 24 types of valid syllogistic reasoning; (2) end-to-end mapping from premises to conclusion introduces contradictory training targets between neural components for pattern recognition and logical reasoning. Taking the Euler Net as a representative supervised neural network, we experimentally illustrate that the limitations are common to all image-input supervised networks. We further challenge the most recent ChatGPTs (GPT-5-nano and GPT-5) to determine the satisfiability of syllogistic statements in four surface forms (patterns): words, double words, simple symbols, and long random symbols, showing that surface forms affect the reasoning performance and that ChatGPT GPT-5 may reach 100% accuracy but still provide incorrect explanations. As empirical training processes are stopped after achieving 100% accuracy, we conclude that supervised machine learning systems may follow scaling laws but will not attain the rigour of symbolic logical reasoning.

## 1 INTRODUCTION

The historical success of neural networks, particularly LLMs, has been witnessed in various applications, such as human-like communication (Biever, 2023), playing games (Silver et al., 2017; Schrittwieser et al., 2020), predicting gene structures (Abramson et al., 2024), and solving mathematical tasks (Davies et al., 2021; Trinh et al., 2024). By increasing the amount of training data and training time (Kaplan et al., 2020; Bahri et al., 2024) and breaking complex tasks into multiple steps (Creswell et al., 2022; Wei et al., 2023; Lightman et al., 2023), data-driven machine learning systems may steadily enhance their reasoning capabilities. However, their reasoning abilities are still limited, even for simple logical reasoning (Biever, 2023), for example, the syllogistic reasoning system (Eisape et al., 2024; Lampinen et al., 2024; Kim et al., 2025), where the reasoning process is primitive and cannot be broken into multiple steps. Recently, by promoting vector embeddings into spheres and introducing the method of *reasoning through explicit model construction and inspection* (Johnson-Laird & Byrne, 1991; Knauff et al., 2003; Goodwin & Johnson-Laird, 2005; Knauff, 2009), Sphere Neural Networks (SphNN) successfully go out of the paradigm of data-driven machine learning and achieve the rigour of symbolic syllogistic reasoning (Dong et al., 2024; 2025). This is not surprising, as RNNs are Turing complete (Nowak et al., 2023; Strobl et al., 2024) and SphNN is a special RNN. However, this raises the question of whether data-driven machine learning systems can reach (or be infinitely close to) the same performance by increasing the amount of training data and training time. Here, we report two methodological limitations that prevent them from achieving symbolic-level syllogistic reasoning; thus, they will not achieve symbolic-level logical reasoning, as syllogistic reasoning is the foundation and a subset of logical reasoning (Khemlani & Johnson-Laird, 2012; Malpass & Marfori, 2017).

This paper is structured as follows: Section 2 introduces the criterion of symbolic-level syllogistic reasoning. Section 3 surveys supervised neural syllogistic reasoning, recent assessments of LLMs in syllogistic reasoning, neural logical proving, and ends with our research question. Section 4 presents two limitations that prevent all data-driven machine learning systems from reaching symbolic-level syllogistic reasoning: (1) Training data cannot distinguish every valid type of syllogistic reasoning; (2) End-to-end mapping introduces contradictory targets between neural components of pattern recognition and logical reasoning. Using Euler Net as a representative image-input supervised neural network for syllogistic reasoning, section 5 shows that composition tables cannot distinguish syllogistic reasonings with the same premises but different conclusions, and what kinds of unintended inputs an end-to-end mapping process will generate. With recent GPT-5-nano and GPT-5, we experimentally demonstrate their unstable performances in syllogistic reasoning across four surface forms: words, double words, simple symbols, and random symbols. Experiments with Euler Net and two GPT versions convergently show that they follow the Scaling Law in increasing syllogistic reasoning performances, but can not achieve the symbolic level. Section 6 concludes the work and lists several research directions.

## 2 SYLLOGISTIC REASONING: THE ORIGIN OF LOGICAL REASONING

The central notion of logical reasoning, from the origin of logic research in history till now, is the notion of "following from", or more formally, "logical consequence from the premises" – what can we know from the premises? Syllogistic reasoning, developed by Aristotle over 2,000 years ago, is the start of the history of logical reasoning. From syllogistic reasoning, logicians developed propositional logic in the Medieval period and first-order logic later (Malpass & Marfori, 2017).

Aristotelian syllogistic reasoning is a deduction with the form of two premises and one conclusion. A syllogistic deduction only contains three terms ($X$, $Y$, and $Z$) and four possible relations: (1) *universal affirmative*: all $X$ are $Y$; (2) *particular affirmative*: some $X$ are $Y$; (3) *universal negative*: no $X$ are $Y$; (4) *particular negative*: some $X$ are not $Y$. Let two premises be *some lawyers are presidents* and *no presidents are scientists*, the conclusion and its negation will be *some lawyers are not scientists* and *all lawyers are scientists*, as shown in Figure 1(e). The four relations can be interpreted through set relations in Euler diagrams, shown in Figure 1(a-d). For example, *some X are Y* can be interpreted as the relation "set X ($\mathcal{O}_X$) intersects with set Y ($\mathcal{O}_Y$)", which corresponds to three possible diagrammatic relations: (1) $\mathcal{O}_X$ partially overlaps with $\mathcal{O}_Y$, (2) $\mathcal{O}_X$ contains $\mathcal{O}_Y$, (3) $\mathcal{O}_Y$ contains $\mathcal{O}_X$. We can merge the three possible relations into one relation: $\mathcal{O}_X$ does not disconnect from $\mathcal{O}_Y$, $\neg\mathbf{D}(\mathcal{O}_X, \mathcal{O}_Y)$, as shown in Figure 1(c). Formally, we define $\mathcal{O}_X$ disconnecting from $\mathcal{O}_Y$ as that there is no $\mathcal{O}_Z$ that is part of $\mathcal{O}_X$ and $\mathcal{O}_Y$.

$$\mathbf{D}(\mathcal{O}_X, \mathcal{O}_Y) \triangleq \nexists \mathcal{O}_Z \mathbf{P}(\mathcal{O}_Z, \mathcal{O}_X) \wedge \mathbf{P}(\mathcal{O}_Z, \mathcal{O}_Y)$$

We can define syllogistic relations through the primitive diagrammatic relation $\mathbf{P}$ (Smith, 1996) and establish a one-to-one mapping ($\Leftrightarrow$) between syllogistic and diagrammatic relations as follows.

- "all $X$ are $Y$" $\Leftrightarrow$ "Circle $\mathcal{O}_X$ is part of Circle $\mathcal{O}_Y$", $\mathbf{P}(\mathcal{O}_X, \mathcal{O}_Y)$;
- "some $X$ are $Y$" $\Leftrightarrow$ "Circle $\mathcal{O}_X$ does not disconnect from Circle $\mathcal{O}_Y$", $\neg\mathbf{D}(\mathcal{O}_X, \mathcal{O}_Y)$;
- "no $X$ are $Y$" $\Leftrightarrow$ "Circle $\mathcal{O}_X$ disconnects from Circle $\mathcal{O}_Y$", $\mathbf{D}(\mathcal{O}_X, \mathcal{O}_Y)$;
- "some $X$ are not $Y$" $\Leftrightarrow$ "Circle $\mathcal{O}_X$ is not part of Circle $\mathcal{O}_Y$", $\neg\mathbf{P}(\mathcal{O}_X, \mathcal{O}_Y)$.

A syllogistic reasoning can be *satisfiable, unsatisfiable, valid*, or *invalid*. Being *satisfiable* means there is a case in which both the premises and the conclusion are true. Being *valid* means the conclusion is true in every case its premises are true (Jeffrey, 1981). For a *valid* reasoning, the negation of its conclusion is *unsatisfiable*; for an *invalid* reasoning, the negation of its conclusion is *satisfiable*. Diagrammatically, syllogistic reasoning is *satisfiable*, if and only if we can construct an Euler diagram, e.g., three circles satisfying the diagrammatic relations of the premises and conclusion; otherwise, this reasoning will be *unsatisfiable*. In Figure 1(g), we successfully constructed an Euler diagram of the premises and the conclusion *some lawyers are not scientists*, so this reasoning is *satisfiable*. But, we cannot construct an Euler diagram of the premises and the conclusion *all lawyers are scientists*, so this conclusion is *unsatisfiable*, and therefore, its negation is *valid*.

If we allow two terms in premises to change positions and fix the order of terms in the conclusion statement, there will be 256 different forms of Aristotelian syllogistic reasoning, among which 24

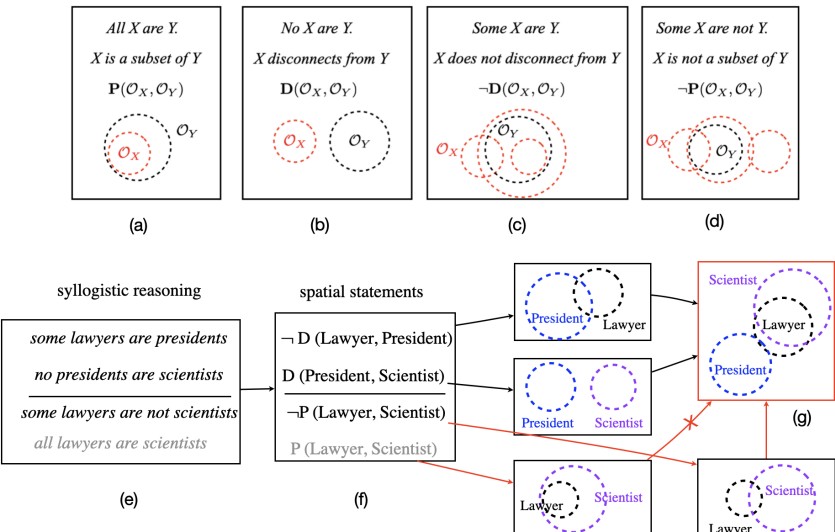

Figure 1: (a-d) Four syllogistic relations and their spatial relations; (e) from the two premises, the logical conclusion is *some lawyers are scientists*, its negation is *all lawyers are scientists*; (f) spatial statements of the syllogistic statements; (g) no sphere configuration satisfies the premises and the conclusion $\mathbf{P}$(lawyers, scientists); there is a sphere configuration that satisfies the premises and the conclusion $\neg\mathbf{P}$(lawyers, scientists)

types (listed in Table 2 in the Appendix) are *valid* (Khemlani & Johnson-Laird, 2012). A reasoning network reaches the rigour of syllogistic reasoning, if it can correctly determine for sure any *valid* syllogistic reasoning and construct counter-examples for *invalid* ones. This criterion also applies to neural networks in reasoning with out-of-distribution data (unintended inputs).

## 3 STATE OF THE ART AND RESEARCH QUESTIONS

As a basic logical deduction, syllogistic reasoning is straightforward for symbolic methods (Vuk-mirovic et al., 2019; Bentkamp et al., 2021). However, developing neural syllogistic models is extremely challenging, to the extent that it was regarded as utopian a decade ago (Khemlani & Johnson-Laird, 2012). The first supervised deep learning for syllogistic reasoning, Euler Net, did not appear until 2018 (Wang et al., 2018), which mapped premises to conclusions, achieving 99.8% accuracy on the benchmark dataset. The large family of data-driven neural networks, Large Language Models (LLMs), can be applied for logical reasoning (Dong & Ma, 2025; Lin et al., 2025; Li et al., 2025), e.g., Goedel-Prover (Lin et al., 2025), the Self-play LLM Theorem Provers (Dong & Ma, 2025). However, in these systems, the correctness of formal states is not determined by LLMs, but by humans or symbolic provers, such as Isabelle, LEAN.

The method of Chain-of-Thought (CoT) (Wei et al., 2023; Li et al., 2025) is a strategy to improve the reasoning performance of neural networks by breaking a task into several intermediate steps, which does not affect the performance of single-step reasoning. Several studies have explored the syllogistic reasoning (as single-step reasoning) performance of LLMs. Eisape et al. (2024) evaluated PaLM 2 family LLMs (Google, 2023) and Llama 2 family LLMs (Touvron et al., 2023), showing that PaLM 2-Small achieved the best accuracy about 75%, better than PaLM 2-Large, which does not strictly follow the Scaling Law. Lampinen et al. (2024) evaluated PaLM 2 LLMs and GPT-3.5 (OpenAI, 2023), concluding that LLMs may achieve above-chance performances in familiar situations but exhibit numerous imperfections in abstract reasoning, including syllogism. Wysocka et al. (2025) examined Mistral LLMs (Jiang et al., 2023; Mistral, 2023), Gemma LLMs (Gemma & Google, 2024), Llama-3 LLMs (MetaAI, 2024)), and BioMistral LLMs (Labrak et al., 2024), with conclusions that zero-shot LLMs achieved an average accuracy between 70% on *generalised modus ponens* and 23% on *disjunctive syllogism*, and both zero- and few-shot LLMs are sensitive to surface-level lexical variations. Thus, they are far from achieving the reliability required for high-

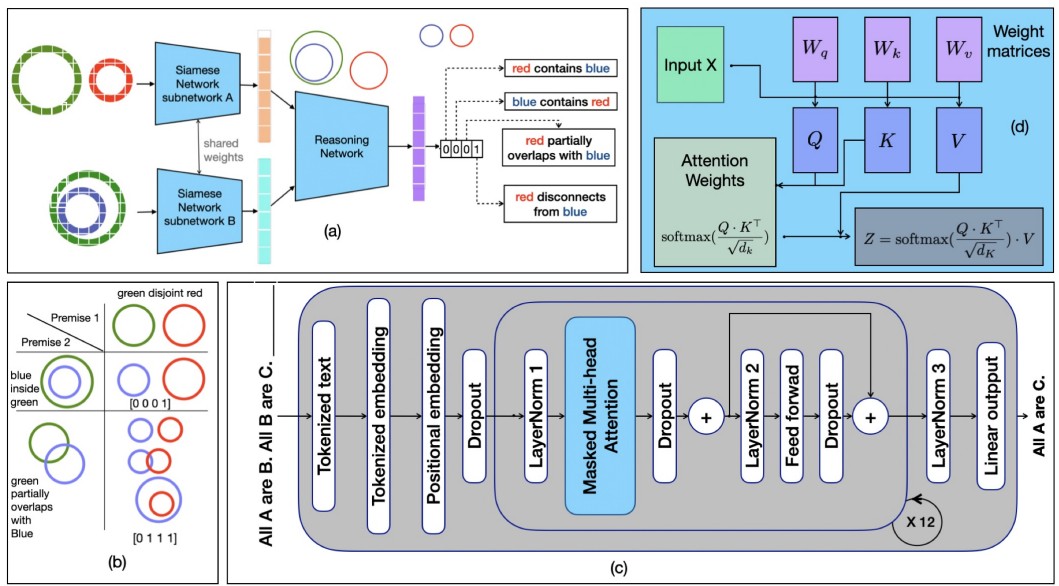

Figure 2: Data-driven machine learning systems for syllogistic reasoning. (a) Supervised deep learning using image inputs, e.g., Euler Net; (b) the composition table that generates training data. Each row and each column are two premises, the intersection cell is the conclusion; (c) a GPT architecture for syllogistic reasoning with symbolic inputs; (d) the attention mechanism of transformers. $Q$, $K$, and $V$ form a statistical combination table.

stakes biomedical applications, let alone attaining the rigour of symbolic-level reasoning. Dong et al. (2025) evaluated GPT-3.5-turbo and GPT-4 in determining the validity of all types of classic syllogistic reasoning in three lexical forms: (1) meaningful words, (2) simple symbols, and (3) long random symbols, showing that ChatGPT (GPT-3.5-turbo) reached the best performance (correct decision and explanation) of $46.9\%$ using statements with simple symbols, and ChatGPT (GPT-4o) reached the best performance of $82.4\%$ with long random symbols.

Considering (1) the scaling law (Kaplan et al., 2020; Bahri et al., 2024), (2) the huge training costs (in terms of data, GPUs, and training time) of LLMs, (3) the Turing Completeness of recurrent neural nets (SIE, 1995; Nowak et al., 2023), and (4) Sphere Neural Networks (Dong et al., 2025), our research question can be stated as follows: Can data-driven neural networks reach or be infinitely close to this level if the amount of training data increases to infinity? A negative answer will lead to the conclusion that supervised neural networks cannot reach the symbolic level of logical reasoning, because syllogistic reasoning is the foundation of logical reasoning.

## 4 METHODOLOGICAL DEFICITS OF DATA-DRIVEN MACHINE LEARNING

In this section, we disclose two methodological deficits that are introduced inevitably by utilising training data, which prevent data-driven neural networks from achieving the symbolic level of syllogistic reasoning. Our analysis will start with Euler Net, the only supervised neural network that reaches 99.8% accuracy for syllogistic reasoning, and extend our conclusions to Transformer architectures and LLMs.

### 4.1 TRAINING DATA CANNOT DISTINGUISH EACH VALID TYPE OF SYLLOGISTIC REASONING

Inspired by the structure of the human visual cortex, Wang et al. (2018; 2020) developed Euler Net, whose inputs are two images, each consisting of two coloured circles with a set-theoretic relation, as illustrated in Figure 2(a). Colours of circles distinguish three terms in syllogistic reasoning. The common colour in the two input images is the mid-term. Two Siamese networks encode each input image into a latent vector. The output of Euler Net is a vector representing the set-theoretic relation(s) between the subject and the predicate. The mapping from two premises to conclusions is

Figure 3: The combination table establishes associations between inputs (premises) and output (conclusion). Premises of *Some ... are (not) ...* occupy three columns or rows. The cell with the green boundary hosts 5 valid types of syllogistic reasoning.

enumerated in the combination table, where conclusions are symbolised as a vector, as illustrated in Figure 2(b). The training data takes the form of ((image, image), vector). The benchmark dataset consists of 96000 pieces of data. They are generated by the combination table as shown in Figure 3. Euler Net achieved 99.8% accuracy on the testing dataset. As shown in Figure 3, the two syllogistic premises *some ... are ...* and *some ... are not ...* occupy three rows and columns. Nine table cells contain various syllogistic conclusions. This follows the fact that training data cannot separate each valid type of syllogistic reasoning. This produces the phenomenon that Euler Net demonstrates close 100% accuracy in the benchmark datasets, but, its performance in determining the correctness for each valid syllogistic reasoning ranges from 50% to 100%, as listed in Table 3 in the experiment. This is the first deficit: through mapping syllogistic premises to conclusions in the form of set-theoretic diagrams, we will not have training data that distinguishes each valid syllogistic reasoning.

## 4.2 END-TO-END LEARNING INTRODUCES CONTRADICTORY TRAINING TARGETS

The architecture of image-input supervised neural networks is an end-to-end pipeline from a pattern recognition component to a reasoning component. The pattern recognition component recognises objects in input images. The reasoning component integrates recognised objects in the two input images into one model and predicts the relation between target objects from the model. A well-trained deep-learning pattern recognition system can recognise an object from its parts, which is a desired feature in Computer Vision (He et al., 2022) – Siamese architecture was used to recover frames in video

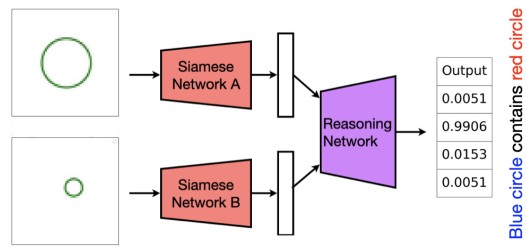

Figure 4: Euler Net may inject blue and red circles into the inputs and predict "blue contains red".

recognition (Gupta et al., 2023), and can recover the whole image of the next frame, given the current image and only 5% of the image of the next frame. However, logical deduction is to identify information implicitly in the premises (Simon, 2019); thus, injecting new objects into the premises is not allowed. This is the second deficit: an end-to-end pipeline that maps the premises to the conclusion introduces contradictory training targets between the neural components of pattern recognition and logical reasoning – the pattern recognition component may inject new objects that do not exist

in the input images, and the reasoning component can neither stop nor notice this. For example, the Siamese networks (pattern recognition components) of Euler Net may inject red and blue circles into the input images, causing the reasoning component to output $[0.0051, 0.9906, 0.0153, 0.0051]$, which means "blue circle contains red circle", while the input images have only two single green circles, as shown in Figure 4.

### 4.3 EXTENDING TO TRANSFORMERS AND LLMS

Any data-driven neural network that suffers from one of the two deficits will not achieve the symbolic level performance in syllogistic reasoning. Here, we extend our analysis to Transformers and Transformer-based LLMs, showing they cannot have perfect training data to distinguish each valid type of syllogistic reasoning (deficit 1); in Experiment 3, we demonstrate LLMs' reasoning performances are affected by lexical patterns (deficit 2).

The basic function LLMs is to complete sentences. For example, given *all A are B and all B are C. Therefore, ____*, a well-trained LLM will complete *all A are C*, as illustrated in Figure 2(c). This is a well-known type (BARBARA) of syllogistic reasoning. Another valid type (BARBARI) has the same premise but a different conclusion (we list all valid types and their names of syllogistic reasoning in the Appendix B): *some A are C*. LLMs are data-driven neural networks whose training data have both types. The centre learning component is the Multihead Attention mechanism (Vaswani et al., 2017), as illustrated in Figure 2(d). The learnable weight matrices $W_q$, $W_k$, $W_v$ automatically learn associations among concepts in linguistic corpora and represent them as a Key-Query-Value table structure: Keys and Queries are organised as rows and columns of a table, and the Values are the cells (Raschka, 2024, pp.70-72). Compared to other data-driven neural networks, such as RNNs and LSTMs, Transformers can establish long-distance associations (Qin et al., 2023), but this does not enable them to distinguish each valid type of syllogistic reasoning.

## 5 EXPERIMENTS

Achieving symbolic-level logical reasoning requires neural networks to make decisions for sure without using symbolic provers. We design experiments to examine whether increasing training data will allow Euler Net (Wang et al., 2018; 2020) to improve its performance infinitely close to the symbolic level.

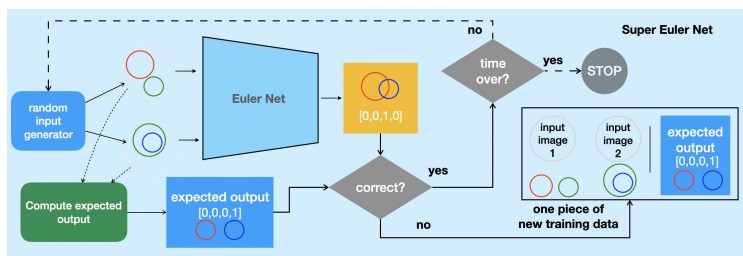

Figure 5: The architecture of Super Euler Net.

In the setting of Euler Net, all image inputs are circles, therefore, the correct outputs of syllogistic reasoning can be computed. We extend Euler Net into Super Euler Net (SupEN) that can automatically identify incorrect output of Euler Net and create new training data, as illustrated in Figure 5: SupEN randomly generates images, and checks whether the output of Euler Net is correct (that is, the binary cross-entropy loss between the network output and the correct output is less than a threshold). If not, a new piece of input-output pair for training will be created. The main procedure is outlined in Algorithm 1 (in the Appendix A).

### 5.1 EXPERIMENT 1

**The aim:** The pattern recognition component of Euler Net may automatically recognise the whole from the parts (Figure 4). If we introduce a new class *unintended inputs* for all the parts, we check whether increasing training data can exhaust the parts.

**Setting of the experiment** We define a new output vector $[0, 0, 0, 0]$ representing unintended inputs and train SupEN to classify single-circled inputs into $[0, 0, 0, 0]$ till it reaches 100% accuracy. In the end, SupEN can perform syllogistic reasoning for regular inputs and classify single-circle inputs as

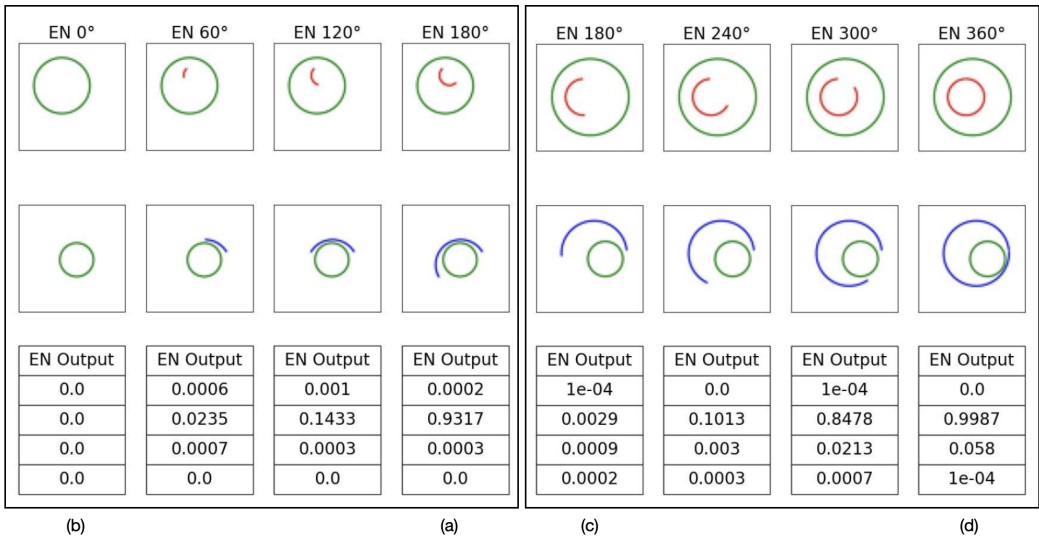

Figure 6: (a) Super Euler Net may automatically complete half circle into full circle (with the output $[0.0002, 0.9317, 0.0003, 0]$). As we decrease the length of the arc to $120°$, $60°$, and $0°$, it decreases this value accordingly. (b) Super Euler Net may automatically ignore the half circle and only take one green circle as input (with the output $[0.0001, 0.0029, 0.0009, 0.0002]$). As we increase the length of the arc to $240°$, $300°$, and $360°$, it increases this value accordingly.

unintended. Then, we create a new dataset, in which one input image consists of a green circle and a half red circle, and the other image consists of a half blue circle and a green circle.

**Experiment results** Experiments show that sometimes SupEN completes the two half circles into two whole circles, and concludes $[0, 1, 0, 0]$ *the blue circle contains the red circle*, as shown in Figure 6(a). In this case, if we decrease the arc length of the two half circles, the confidence value flagging *the blue circle contains the red circle* will decrease, as shown in input images and outputs from Figure 6(a) to (b). Sometimes, SupEN simplified one green circle and a half circle into one green circle (half circles are neglected) and concludes the inputs are unintended $[0, 0, 0, 0]$, as shown in Figure 6(c). In this case, if we increase the arc length of the two half circles, the confidence value flagging *the blue circle contains the red circle* will increase correspondingly, as shown in input images and outputs from Figure 6(c) to (d). This, however, will automatically create another kind of unintended pattern: one green circle and a partial circle with $(180° + 360°)/2 = 270°$. This loop will never end.

**Conclusion** Training data can not exhaust unintended inputs, for new training data generates new unintended inputs. Thus, SupEN will not reach symbolic-level reasoning if we do not restrict inputs.

## 5.2 EXPERIMENT 2

**The aim** If we restrict all inputs to be intended (either two complete circles or one single circle) and repeatedly increase training data, we check whether SupEN will increase performance and be *infinitely close* to the symbolic level.

**Setting of the experiment** As SupNN can automatically identify reasoning errors and generate new training data, we let it repeatedly switch between searching for new training data and training using new training data. In the search procedure, the central point and the radius of a circle are random, with two restrictions as follows: (1) circles are fully inside the boundary of an image; (2) the minimum radius is 0.1. We allow all possible combinations between two circles. Following these criteria, SupEN creates a new testing data set $\mathcal{D}_1$.

In the training stage, SupEN creates a training data $\mathcal{D}_T$ that consists of newly created training data $\mathcal{D}_1$ and original training data $\mathcal{D}_2$, $\mathcal{D}_T = \mathcal{D}_1 \cup \mathcal{D}_2$, as shown in Figure 7. For example, $\mathcal{D}_1$ is a newly created dataset with one-circle images. The size of $\mathcal{D}_2$ is 9 times larger than that of $\mathcal{D}_1$.

**Tesing with random dataset** Different from the standard deep learning paradigm, in which testing data and training data shall have the same distribution (Goyal & Bengio, 2022), we need to evaluate whether SupEN can reach (or be infinitely close to) 100% accuracy for new testing data. Thus, in this experiment, the testing data are randomly generated. We let SupEN loop 20 times through the searching-training process to improve its reasoning performance.

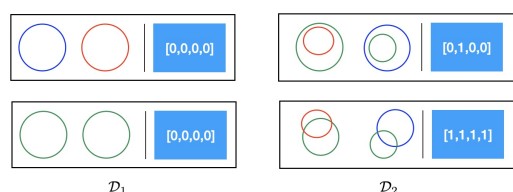

Figure 7: $\mathcal{D}_1$ is a newly created dataset; $\mathcal{D}_2$ is randomly selected from the standard training dataset.

**Experiment result** Being tested with randomly generated testing data, SupEN reaches $56.0\%$, before the loop (This SupEN corresponds to a well-trained Euler Net). Through repeated training of newly created datasets, the accuracy improves steadily and reaches a peak value of $97.8\%$ after the 19-th iteration, as illustrated in Figure 8. The increase of accuracy support the law of scaling. The oscillation is because we randomly search instead of searching with gradual descent operations.

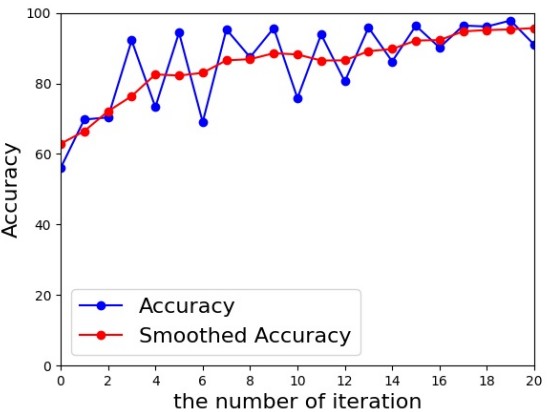

Figure 8: The accuracy of SupEN at each iteration with newly generated training data.

**Tesing with data covering all valid types of syllogistic reasoning** Reaching symbolic-level syllogistic reasoning requires to reach or be infinitely close to 100% accuracy for each valid type of syllogistic reasoning. Thus, we created another testing dataset covering all 24 valid types as follows: We group equivalent syllogistic statements into the same group, for example, *no x are y* and *no y are x* are in the same group. This reduces 24 *valid* syllogism types into 14 groups. For each group, we created 500 different premises by extracting hypernym relations from WordNet-3.0 (Miller, 1995). For each premise, we deduce its valid logical conclusion and its negation, totalling 14000 syllogism reasoning tasks. In the hypernym structure, *elementary_particle.n.01* is a descendent of *natural_object.n.01* and *artifact.n.01* is not a descendent of *natural_object.n.01*. So, we create the valid syllogistic reasoning and its negation, as shown below. We use the pre-processing tool of the original Euler Net to transform premises into coloured circles, and conclusions into vectors.

**Experiment result** We fed the new dataset to a well-trained SupEN (with 19-th loop of improvements). It works very well if a task falls into a valid syllogistic structure: For 8 syllogistic structures, it reaches 100% accuracy, namely, BAR-BARA, CELARENT, CESARE, DARA-

*all elementary_particle.n.01 are natural_object.n.01.*
*no artifact.n.01 are natural_object.n.01.*

*no elementary_particle.n.01 are artifact.n.01.* ∴

*all elementary_particle.n.01 are natural_object.n.01.*
*some artifact.n.01 are natural_object.n.01.*

*some elementary_particle.n.01 are artifact.n.01.* ∴

PTI, CALEMES, CAMESTRES, FELAPTON, and FESAPO. Accuracies of the remaining 16 types range from $50\%$ to $83.3\%$. The overall accuracy is 76%, as shown in Table 3 (in the Appendix C). This performance is consistent with Eisape et al. (2024)'s evaluation with PaLM 2 and Llama 2 family LLMs — the best performance was achieved by PaLM 2-Small with accuracy about 75%, better than PaLM 2-Large. These results suggest that the reasoning performance may not be infinitely close to the symbolic level solely by increasing training data and training time (in terms of the number of loops).

Table 1: Syllogistic reasoning performance of OpenAI GPT-5-nano and GPT-5. '✓EXPL' for correct explanation, '✗ H' for hallucinative explanation. The '#correct decision-✗ H' column means a correct decision with a wrong explanation; the '#wrong decision-✓EXPL' column means a wrong decision with a correct explanation.

| version | surface form | #correct decision | | #wrong decision | | #simple acc |
|---|---|---|---|---|---|---|
| | | ✓EXPL | ✗ H | ✓EXPL | ✗H | |
| GPT-5-nano | words | 160 ( 62.5%) | 90 | 5 | 1 | 97.7% |
| | double words | **230 ( 89.4%)** | 22 | 4 | 0 | 98.4% |
| | simple symbols | 226 ( 88.3%) | 24 | 6 | 0 | 97.7% |
| | long random symbols | 222 ( 86.7%) | 25 | 9 | 0 | 96.5% |
| GPT-5 | words | **239 ( 93.4%)** | 16 | 1 | 0 | 99.6% |
| | double words | 234 ( 91.4%) | 21 | 1 | 0 | 99.6% |
| | simple symbols | 236 ( 92.2%) | 15 | 5 | 0 | 98.0% |
| | long random symbols | 231 ( 90.2%) | 25 | 0 | 0 | **100.0%** |

## 5.3 EXPERIMENT 3

**The aim**   We evaluate two versions of the most recent Open AI GPTs, GPT-5 and GPT-5-nano, in syllogistic reasoning, to examine whether the scaling law may guarantee the performance to reach (or be infinitely close to) the symbolic-level. Concretely, we examine whether surface lexical patterns can affect their reasoning performance (deficit 2).

**Setting of the experiment**   We follow the method in (Dong et al., 2025) that used syllogistic statements with four surface lexical patterns: (1) meaningful words, e.g. *Socrates*, (2) doublele words, e.g. *City_Socrates*, (3) simple symbols, e.g. *S*, and (4) long random symbols, e.g. *VnWKvqcBsEh1*, to determine the satisfiability of all 256 types of classic Aristotelian syllogistic reasoning. The motivation for introducing the new pattern of double words is to enable the meaning of words to support reasoning, while reducing the bias inherent in single words.

**Evaluation metrics**   We use two evaluation metrics: (1) the normal metrics in terms of accuracy (the #simple acc column in Table 1); (2) the metrics of reaching symbolic-level logical reasoning, namely, a response is correct if both the decision and the explanation are correct (the #correct decision-✓EXPL column in Table 1).

**Results**   The experiment's results, measured in normal metrics, range from 97.7% to 100.0%, confirming the high performance of both OpenAI's GPT-5-nano and GPT-5 in syllogistic reasoning. Meanwhile, eight experiments show that surface lexical patterns can affect reasoning performance, and each version made at least 15 correct decisions with incorrect explanations. Being fed with double-word statements, GPT-5-nano achieved 89.4% correct decisions with correct explanations, better than using other forms. With single-word statements, GPT-5 achieved 93.4% correct decisions with correct explanations, better than using other forms. In particular, GPT-5 achieved 100% correct decisions with long random symbols, but 25 correct decisions were supported by wrong explanations. As 100% accuracy is the maximum performance guided by the scaling law, usually accompanied by the stop of training, reaching symbolic-level performance will be beyond this limit.

## 6 CONCLUSION AND OUTLOOKS

Considering the simple forms of syllogistic reasoning, we may assume that deep neural networks can easily solve them. Recent evaluations of neural syllogistic reasoning have focused on linguistic-input neural networks, e.g., LLMs, and explored their internal mechanisms, with the implicit conclusion that they do not reach the symbolic level of syllogistic reasoning. Complementarily, we identify two methodological limitations that prevent supervised neural networks from achieving symbolic-level syllogistic reasoning, although their performance may improve. Our work focuses on the training data and their surface forms – a precondition that all supervised neural networks encounter, suggesting the need to seek alternative neural representations, methods, and architectures to achieve more cost-effective, high-performance, and interpretable neural logical reasoning.

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

## A  THE MAIN PROCEDURE OF SUPER EULER NET

---

**Algorithm 1:** Automatic generating new training data

---

**Input:** Euler Net: EN;
The maximum size of unintended data set: maxSize;
The timer: Timer;
The maximum time: maxTime;
The threshold to be unintended: Threshold
**Output:** A new training data: newData
newData $\leftarrow \emptyset$;
DataSize $\leftarrow 0$;
Timer $\leftarrow$ set_timer();
**while** DataSize $<$ maxSize $\wedge$ Timer $<$ maxTime **do**
   Input $\leftarrow$ randomly_generate_one_input ()
   ENOutput $\leftarrow$ output_of_network (EN, Input);
   Output $\leftarrow$ compute_correct_output (Input)
   **if** loss(ENOutput, Output) $>$ Threshold **then**
      newData $\leftarrow$ newData $\cup$ {(Input, Output)}
      DataSize $\leftarrow$ DataSize $+ 1$

**return** newData

---

## B  THE LIST OF 24 VALID TYPES OF SYLLOGISTIC REASONING

Table 2: List of all 24 valid syllogisms, each having a name whose vowels indicate types of moods, e.g., vowels in 'CELARENT' indicates *universal negative* (E), *universal affirmative* (A), and *universal negative* (E).

| Num | Name | Premise | Conclusion |
|-----|------|---------|------------|
| 1 | BARBARA | all $X$ are $Y$, all $Y$ are $Z$ | all $X$ are $Z$ |
| 2 | BARBARI | all $X$ are $Y$, all $Y$ are $Z$ | some $X$ are $Z$ |
| 3 | CELARENT | no $Y$ is $Z$, all $X$ are $Y$ | no $X$ is $Z$ |
| 4 | CESARE | no $Z$ is $Y$, all $X$ are $Y$ | no $X$ is $Z$ |
| 5 | CALEMES | all $Z$ are $Y$, no $Y$ is $X$ | no $X$ is $Z$ |
| 6 | CAMESTRES | all $Z$ are $Y$, no $X$ is $Y$ | no $X$ is $Z$ |
| 7 | DARII | all $Y$ are $Z$, some $X$ are $Y$ | some $X$ are $Z$ |
| 8 | DATISI | all $Y$ are $Z$, some $Y$ are $X$ | some $X$ are $Z$ |
| 9 | DARAPTI | all $Y$ are $X$, all $Y$ are $Z$ | some $X$ are $Z$ |
| 10 | DISAMIS | some $Y$ are $Z$, all $Y$ are $X$ | some $X$ are $Z$ |
| 11 | DIMATIS | some $Z$ are $Y$, all $Y$ are $X$ | some $X$ are $Z$ |
| 12 | BAROCO | all $Z$ is $Y$, some $X$ are not $Y$ | some $X$ are not $Z$ |
| 13 | CESARO | no $Z$ is $Y$, all $X$ are $Y$ | some $X$ are not $Z$ |
| 14 | CAMESTROS | all $X$ are $Y$, no $Y$ is $Z$ | some $X$ are not $Z$ |
| 15 | CELARONT | no $X$ is $Y$, all $Z$ are $Y$ | some $X$ are not $Z$ |
| 16 | CALEMOS | all $Z$ are $Y$, no $Y$ is $X$ | some $X$ are not $Z$ |
| 17 | BOCARDO | some $Y$ are not $Z$, all $Y$ are $X$ | some $X$ are not $Z$ |
| 18 | BAMALIP | all $Y$ are $X$, all $Z$ are $Y$ | some $X$ are $Z$ |
| 19 | FERIO | some $X$ are $Y$, no $Y$ is $Z$ | some $X$ are not $Z$ |
| 20 | FESTINO | some $X$ are $Y$, no $Z$ is $Y$ | some $X$ are not $Z$ |
| 21 | FERISON | some $Y$ are $X$, no $Y$ is $Z$ | some $X$ are not $Z$ |
| 22 | FRESISON | some $Y$ are $X$, no $Z$ is $Y$ | some $X$ are not $Z$ |
| 23 | FELAPTON | all $Y$ are $X$, no $Y$ is $Z$ | some $X$ are not $Z$ |
| 24 | FESAPO | all $Y$ are $X$, no $Z$ is $Y$ | some $X$ are not $Z$ |

## C    REASONING PERFORMANCES OF EULER NET AFTER 20 TIMES LOOP

Table 3: Performances of SupEN (after 19 loops of improvements) for each valid type of syllogistic reasoning.

| Valid Type | Accuracy | Valid Type | Accuracy | Valid Type | Accuracy |
|------------|----------|------------|----------|------------|----------|
| BARBARA | 100% | BARBARI | 50% | BAROCO | 66.7% |
| BAMALIP | 50% | BOCARDO | 75% | CALEMES | 100% |
| CAMESTROS | 50% | CELARENT | 100% | CESARO | 50% |
| CALEMO | 50% | CESARE | 100% | CELARONT | 50% |
| DARAPTI | 100% | DARII | 75% | DISAMIS | 75% |
| FESAPO | 100% | DATISI | 75% | DIMATIS | 75% |
| FELAPTON | 100% | FERIO | 83.3% | FERISON | 83.3% |
| CAMESTRES | 100% | FRESISON | 83.3% | FESTINO | 83.3% |

## D    CODE AND DATA

https://anonymous.4open.science/r/EN_GPT5-1051/README.md

