# OpenReview forum: "Can Data-driven Machine Learning Reach Symbolic-level Logical Reasoning? -- The Limit of the Scaling Law"
_ICLR.cc/2026/Conference — Submitted to ICLR 2026_

### Official Review · Reviewer_34FN · 2025-10-22

**Soundness:** 2
**Presentation:** 2
**Contribution:** 2
**Rating:** 2
**Confidence:** 3

**Summary:**

The paper addresses whether data-driven neural networks can ever attain symbolic-level logical reasoning by simply scaling up data and computing. Using syllogistic reasoning as the test case, the authors identify two methodological limitations that they claim make this impossible:
(1) training data cannot distinguish between all valid syllogistic reasoning types, and
(2) End-to-end mappings in neural architectures introduce contradictory objectives between pattern recognition and reasoning.

They demonstrate these limitations using the Euler Net and a modified version (Super Euler Net), as well as an evaluation of GPT-5 and GPT-5-nano on symbolic syllogisms across different input forms (words, symbols, random strings). Their experiments show that even high-performing models can reach 100% accuracy but still produce incorrect explanations, which the authors interpret as evidence of an inherent boundary in scaling laws for reasoning.

**Strengths:**

- The topic is ambitious and philosophically interesting, touching on the limits of scaling laws and the intersection between data-driven learning and symbolic reasoning.
- The paper offers a systematic empirical investigation combining vision-based reasoning (Euler Net) and text-based reasoning (GPT models) under a unifying logical framework.
- The focus on syllogistic reasoning is appropriate for testing elementary logical inference, a well-understood and bounded task.
- The experiments are well-organized, and the claims are clearly structured around two proposed methodological deficits.
- The connection to both historical and modern theories of reasoning (Johnson-Laird, Knauff, symbolic logic traditions) adds conceptual depth.

**Weaknesses:**

Framing misalignment.
The title and introduction suggest that the paper addresses the general question of out-of-distribution (OOD) reasoning limitations in data-driven systems. However, the empirical work concerns a very specific and diagrammatic form of syllogistic reasoning. The chosen paradigm introduces additional confounds—such as visual representation and image-based processing—that are orthogonal to the general OOD reasoning question. The findings, therefore, do not substantively advance our understanding of why modern large models fail at OOD reasoning.

Conceptual overreach.
While the identified limitations (non-distinguishable training cases and conflicting submodule objectives) are valid observations, they are not unique to the presented setup and have been documented extensively in prior work (e.g., Guzman et al., Findings of NAACL 2024, and subsequent arXiv extensions). The novelty lies mainly in the framing, rather than the underlying insights. The claim that these two deficits constitute a fundamental limit on all data-driven reasoning systems is too strong, given the narrow empirical base.

Diagrammatic reasoning introduces confounds.
The use of Euler diagrams as the central experimental representation makes the results difficult to generalize. Diagrammatic reasoning blends visual recognition and logical inference, and the errors observed may stem as much from visual encoding issues as from reasoning failures. A purely symbolic or formal logical setup would provide a cleaner testbed for claims about reasoning limitations.

Weak connection to modern architectures.
The paper’s experimental systems (Euler Net and Super Euler Net) are structurally far from state-of-the-art architectures used for large-scale reasoning. Extending results from these simplified models to all “data-driven systems” is not justified without stronger empirical or theoretical bridges.

**Questions:**

Why is a diagrammatic syllogistic reasoning task an appropriate paradigm for investigating general reasoning limits in neural systems?

How would your conclusions differ if the experiments were conducted in a symbolic (non-visual) setup?

Could your two proposed deficits (non-distinguishability and contradictory objectives) be mitigated through architectural or representational interventions?

How do your findings extend—or differ—from existing analyses of reasoning depth and compositionality limitations (e.g., Guzman et al., 2024)?

How do you reconcile the visual-logic mixture of Euler Nets with claims about purely linguistic models like GPT-5?

---

> ### Author Response · Authors · 2025-11-16
>
> Thank you for the comments from a different perspective.
>
> 1. Why is a diagrammatic syllogistic reasoning task an appropriate paradigm for investigating general reasoning limits in neural systems?
>
> Unlike your perspective and that of Guzman et al. (2024), we take a developmental perspective to examine the reasoning of animals, humans, and machines. We focus on Aristotelian syllogistic reasoning, which was the start of the history of logical reasoning, the intersection of the psychological research of animal and human reasoning. Monkeys can do syllogistic reasoning, but they cannot speak like us. If artificial neural networks simulate the functions of biological neural networks, they should be able to perform basic Aristotelian syllogistic reasoning, first at the semantic level via visual inputs, later at the syntactic level via symbolic linguistic inputs. Linguistic symbols and syntax are built upon semantic models. Diagrammatic syllogistic reasoning is syllogistic reasoning at the semantic level. In the psychological literature, these diagrams are called “mental models”. Mental model theory is the most promising theory to explain human syllogistic reasoning. Therefore, diagrammatic syllogistic reasoning is the appropriate paradigm for investigating the general reasoning of humans and of artificial neural systems.
>
> 2. How would your conclusions differ if the experiments were conducted in a symbolic (non-visual) setup?
>
> That will be no difference, or even worse. Because in the symbolic setting, a neural network achieving the symbolic level of syllogistic reasoning “all X are Y, all Y are Z, therefore all X are Z” means this neural network shall conclude this reasoning being valid for any three symbols, X, Y, and Z. The performance will drop when the inputs contain rare symbols. Existing experiments shall that the best performance of LLMs in syllogistic reasoning is around 75% accuracy.
>
> 3. Could your two proposed deficits be mitigated through architectural or representational interventions?
>
> Yes. We referenced Sphere Neural Networks (Dong et al., 2024/2025), which achieve symbolic-level syllogistic reasoning, which successfully void the two deficits as follows: (1) they do not use training data, (2) they extend vector embeddings into sphere embeddings and explicitly represent set-theoretic semantics (different representation), (3) they solve syllogistic reasoning by constructing semantic models (different architecture).
>
> 4. How do your findings extend—or differ—from existing analyses of reasoning depth and compositionality limitations (e.g., Guzman et al., 2024)?
>
> The work of Guzman et al. (2024) is good and is at the syntax level. It analysed the reasoning depth and compositionality limitations of neural networks. Our work shows that supervised neural networks has limitations in solving atomic syllogistic reasoning. Therefore, no conflict; instead, Guzman et al. (2024) and our work together can make a complete story.
>
> 5. How do you reconcile the visual-logic mixture of Euler Nets with claims about purely linguistic models like GPT-5?
>
> Logical reasoning has two facets: syntax and semantics. The inputs of Euler Net represent the semantics, while the inputs of GPT-5 are at the syntax level.  “Visual-logic mixture” is an alternative term for “visual spatial representation” in the psychological literature, which triggers the construction of mental spatial models. Mental spatial models support spatial reasoning and also abstract reasoning in language.  Purely linguistic models like GPT-5, if they genuinely have a human-like understanding of the language, shall have spatial semantics encoded in latent vectors.
>
> 6. Conceptual overreach and Weak connection to modern architectures.
>
> One of the novelties of our work is to examine the limitations of using data to train supervised deep learning for logical reasoning. There is a metaphor in the community of data science and machine learning: data is fuel, and deep learning neural networks are rockets. We show that there is no perfect fuel for the rocket to fly as high as it wants, due to two deficits. We claim any supervised neural networks with the deficits will not reach the symbolic level of rigorous reasoning. Thus, we do not claim that the two deficits constitute a fundamental limit on all neural networks. And we are happy to see that modern neural networks, Sphere Neural Networks, successfully avoid the two deficits and have reached the symbolic level of rigorous syllogistic reasoning.
>
> We will reference Guzman et al. (2024) in our paper and clarify the different perspectives and focuses, and emphasise that the two deficits constitute a fundamental limit on those data-driven neural networks that use training data for end-to-end mapping.

---

> > ### Comment · Reviewer_34FN · 2025-11-25
> >
> > Thank you for your explanations. Perhaps I have been a bit too critical of the paper. I will update the score slightly.

---

> > > ### Author Response · Authors · 2025-11-27
> > >
> > > Dear Reviewer,
> > >
> > > We’re glad our explanation clarified the issue, and we appreciate your willingness to update the score.

---

### Official Review · Reviewer_T9nM · 2025-10-27

**Soundness:** 3
**Presentation:** 2
**Contribution:** 2
**Rating:** 4
**Confidence:** 4

**Summary:**

This paper investigates the symbolic-level logical reasoning limits of data-driven machine learning systems, focusing on supervised deep learning and large language models (LLMs). The authors focus on syllogistic reasoning as a test case for whether neural networks can attain symbolic-level rigor.

Through experimental analyses, the paper identifies two key limitations in data-driven systems:

- Training data cannot distinguish all valid types of syllogistic reasoning.
- End-to-end learning introduces contradictory targets between the pattern recognition and reasoning components.

Using the Euler Net and its improved version, Super Euler Net (SupEN), the authors empirically demonstrate that even with iterative data augmentation and scaling, the model’s accuracy saturates below symbolic-level performance. Extending the argument to Transformer-based LLMs, including GPT-5 and GPT-5-nano, they find that although these models can achieve nearly perfect accuracy, they still produce logically incorrect explanations. The paper concludes that scaling laws allow for performance improvements but do not enable symbolic-level logical reasoning.

**Strengths:**

-  The paper tackles a critical research topic, whether data-driven machine learning can replicate symbolic reasoning.
-  The experiments with Super Euler Net, and GPT-5 variants support the paper’s claim

**Weaknesses:**

-  The experimental scope is narrow, with neural network methods limited exclusively to (Super) EulerNet.
-  the claim "we conclude that supervised machine learning systems may follow scaling laws but will not attain the rigour of symbolic logical reasoning." is overgeneralized

**Questions:**

First of all, I would like to thank the authors for their work. I agree with the authors that large language models (LLMs) and most deep neural networks currently lack robust logical reasoning capabilities. However, I respectfully disagree with the claim: “we conclude that supervised machine learning systems may follow scaling laws but will not attain the rigour of symbolic logical reasoning.”

This statement feels somewhat overgeneralized and represents a limitation of the paper. The experimental scope is restricted to (Super) EulerNet as the sole example of neural network-based reasoning. However, several other architectures have been proposed that demonstrate varying degrees of logical reasoning ability, such as Neural Module Networks [1], Logic Tensor Networks [2], and differentiable inductive logic programming (ILP) models [3]. These approaches are not discussed or compared in the current work.

Would the authors consider clarifying how these methods differ from the approach proposed in the paper, or including such models as additional baselines, or at least discussing their relevance in the broader context of neural reasoning?

[1] Andreas, Jacob, et al. "Neural module networks." Proceedings of the IEEE conference on computer vision and pattern recognition. 2016.

[2] Badreddine, Samy, et al. "Logic tensor networks." Artificial Intelligence 303 (2022): 103649.

[3] Evans, Richard, and Edward Grefenstette. "Learning explanatory rules from noisy data." Journal of Artificial Intelligence Research 61 (2018): 1-64.

---

> ### Author Response · Authors · 2025-11-16
>
> 1. However, I respectfully disagree with the claim: “we conclude that supervised machine learning systems may follow scaling laws but will not attain the rigour of symbolic logical reasoning.” This statement feels somewhat overgeneralized and represents a limitation of the paper.
>
> Thank you for your comment. Research on using neural networks to simulate reasoning can proceed in one of two opposite directions. The popular direction is constructive, with increasing complexity, from perceptron to MPL, then to CNN and RNN, from RNN to LSTM and Transformer, LLMs. The other direction, which is not so familiar, proceeds, by analysing, to greater abstractness – instead of asking what we can construct using existing neural components, we ask instead what are the invariant logical features that are preserved during the construction. It is the fact of pursuing this less familiar opposite direction that characterises this paper.
>
> We abstract two features from the Euler Net, which cause it not to be able to reach the symbolic level rigour of syllogistic reasoning. Our analysis shows that the two features are invariant across the construction of more complex neural network systems.
>
> If artificial neural networks simulate the functions of biological neural networks, they should be able to perform basic Aristotelian syllogistic reasoning at the semantic level first via visual inputs, and later at the syntactic level via symbolic linguistic inputs. Here, we consider whether pure supervised neural networks can reach the symbolic level rigor of syllogistic reasoning.
>
> 2. Would the authors consider clarifying how these methods differ from the approach proposed in the paper, or including such models as additional baselines, or at least discussing their relevance in the broader context of neural reasoning ? [1] Andreas, Jacob, et al. "Neural module networks." Proceedings of the IEEE conference on computer vision and pattern recognition. 2016. [2] Badreddine, Samy, et al. "Logic tensor networks." Artificial Intelligence 303 (2022): 103649. [3] Evans, Richard, and Edward Grefenstette. "Learning explanatory rules from noisy data." Journal of Artificial Intelligence Research 61 (2018): 1-64.
>
> The neural networks in the three papers are neurosymbolic systems, instead of  pure supervised neural networks.
>
> Neural Module Networks (NMN) [1] are neurosymbolic systems for Visual Question-Answering. NMN decomposes a question into symbolic substructures. The function of each symbolic substructure is approximated by supervised neural networks, CNN, FC, LSTM, to inspect a visual image or give answers.
>
> Logic Tensor Networks (LTN) [2] integrate symbolic reasoning with neural networks. LTN allows for end-to-end differentiable learning of first-order logical rules and extends true-false value into degrees of truth. This can handle uncertainties and noise in data, but loses the chance to reach the rigour of symbolic reasoning.
>
> Learning explanatory rules from noisy data [3] combined the strengths of neural networks and logic programming. It applies logic programming to derive rules from data and uses neural networks to handle noise. The target of [2,3] is not to increase the reasoning ability of artificial neural networks.
>
> Despite the practical usefulness of [1,2,3], the theoretical challenge for artificial neural networks remains: if they are the computational model of our biological neural networks that can do rigorous logical reasoning, why, till today, can they, being trained by almost all the data, still not reach the rigour of syllogistic reasoning? The syllogistic premises have all the information to determine the validity of a syllogistic conclusion, why should we believe that using more external data can improve the reasoning performance? In this sense, the scaling law shall be irrelevant to reasoning performance. On the other hand, recent Sphere Neural Networks [4,5] have been proven to reach the symbolic level of syllogistic reasoning without using training data and can be extended to various forms of logical reasoning.
>
> [4] Tiansi Dong, Mateja Jamnik, Pietro Liò (2025). Neural Reasoning for Sure Through Constructing Explainable Models. AAAI.
>
> [5] Tiansi Dong, Mateja Jamnik, Pietro Liò (2024). Sphere Neural Networks for Rational Reasoning. ArXiv:2403.15297.
>
> Therefore, our claim is not overgeneralized. We would take a humble attitude to emphasize our unpopular methodology and the methodological mismatch between the scaling law and basic logical deduction.

---

### Official Review · Reviewer_Vfhs · 2025-10-30

**Soundness:** 3
**Presentation:** 3
**Contribution:** 3
**Rating:** 6
**Confidence:** 4

**Summary:**

This paper explores a fundamental question: Can data-driven machine learning systems—especially supervised neural networks and large language models (LLMs)—achieve symbolic-level logical reasoning solely by scaling data and computational resources?
The authors approach this question by analyzing two representative paradigms: Euler Net, a vision-based supervised network for syllogistic reasoning; and GPT-5 / GPT-5-nano, large-scale transformer-based LLMs.
They identify two methodological limitations that fundamentally constrain data-driven systems: Training data cannot distinguish all valid types of syllogistic reasoning, because different syllogisms may share identical surface patterns in data space, preventing the model from learning symbolic distinctions; End-to-end architectures introduce internal conflicts between pattern recognition (which tends to fill in missing or perceptually inferred information) and logical reasoning (which must not introduce unseen entities).
To empirically verify these hypotheses, the paper develops Super Euler Net, an enhanced version capable of self-generating and validating new training data. Through iterative experiments, it shows that while performance can approach but never surpass 97–100% accuracy, models fail to reach the symbolic rigor required for logical validity and explanation consistency.
The evaluation of GPT-5 further supports this claim: though achieving perfect accuracy in some conditions, the models still produce logically incorrect or hallucinative explanations. The paper concludes that while scaling laws effectively enhance performance, they cannot bridge the gap between data-driven pattern learning and formal logical reasoning.
Overall, this paper provides both theoretical and experimental insights into the limits of scaling in achieving symbolic reasoning and contributes to a broader understanding of how machine learning interacts with formal logic.

**Strengths:**

1) The paper tackles one of the most fundamental open questions in AI—whether data scaling alone can yield true reasoning capabilities. By analyzing the structural mismatch between empirical learning and formal logic, it provides a well-articulated theoretical contribution with clear philosophical and computational implications. Unlike many conceptual discussions, this paper grounds its arguments in experimental evidence. The combination of the Super Euler Net (for visual reasoning) and GPT-5 experiments (for linguistic reasoning) effectively bridges symbolic, perceptual, and neural domains.
2) The two identified deficits—data indistinguishability and end-to-end conflict—are general and apply to a broad class of neural architectures, including both CNN-based and transformer-based systems. This insight is highly valuable for the theory of neural reasoning.
3) The introduction of Super Euler Net is innovative: it automatically generates and validates new training data, providing an interpretable testing ground for studying the scaling behavior of logical inference systems. It also offers a replicable way to study symbolic consistency in vision-based reasoning.
4) The experiments with GPT-5 demonstrate an important phenomenon: models may reach near-perfect accuracy yet still fail to produce correct logical explanations. This observation highlights the distinction between statistical competence and logical understanding.
5) The paper successfully combines perspectives from computer vision, formal logic, and cognitive science. It connects deep learning scaling theory (e.g., Kaplan et al., Bahri et al.) with cognitive models of reasoning (e.g., Johnson-Laird, Knauff), thus contributing to a richer theoretical landscape.

**Weaknesses:**

1) The study focuses exclusively on syllogistic reasoning, which, while fundamental, represents only a small subset of logical reasoning tasks. It remains unclear whether the same limitations would hold for more complex forms such as propositional, predicate, or modal logic. The claim of “methodological impossibility” thus feels somewhat overstated given the narrow scope.
2) The experiments rely on controlled and synthetic inputs (circles, hypernym pairs, etc.), which may not reflect the complexity or ambiguity of real-world reasoning contexts. Extending the analysis to noisy or real-world data could provide stronger evidence for generalization.
3) The evaluation of GPT-5 and GPT-5-nano, while illustrative, limits reproducibility and transparency. Without open-source benchmarks, it is difficult to verify the validity of the reported results or to replicate the symbolic explanation failures.
4) While qualitative analysis is thorough, the statistical treatment of results (e.g., variance across trials, significance testing) is absent. Including such analyses would make the claims more robust and empirically grounded.
5) The conclusion that “scaling laws cannot reach symbolic-level reasoning” is philosophically appealing but empirically underdetermined. It might be more accurate to phrase it as a “current limitation” rather than a “fundamental impossibility,” leaving room for potential hybrid or unsupervised solutions.

**Questions:**

1) Can Super Euler Net or the broader analytical framework be extended to address multi-step or relational reasoning tasks such as first-order logic proofs or causal inference? How might the identified deficits manifest in these more complex settings?
2) Have you explored the possibility of combining your approach with formal symbolic provers (e.g., Goedel-Prover, Isabelle, or LEAN)? Such hybrid systems might circumvent the “end-to-end contradiction” by separating perception from logical validation.
3) Could non-supervised objectives, such as contrastive learning or RL-based symbolic verification, mitigate the issues caused by incomplete training data or pattern injection? Would this change your conclusion about the universality of the deficits?
4) Since syllogistic reasoning can be expressed as triplet relations (subject–predicate–object), have you considered representing reasoning structures using graph neural networks or relational transformers? These may inherently encode logical compositionality and offer a path toward symbolic consistency.
5) To what extent do you view your conclusion (“data-driven ML cannot reach symbolic reasoning”) as a theoretical impossibility versus a practical limitation? Clarifying this distinction could prevent misinterpretation and better position your work within the ongoing debate on neural-symbolic reasoning.

---

> ### Author Response · Authors · 2025-11-16
>
> 1. The study focuses exclusively on syllogistic reasoning, .. It remains unclear whether the same limitations would hold for more complex forms ... The claim of “methodological impossibility” thus feels somewhat overstated given the narrow scope.
>
> Thanks for the great comment. Syllogistic reasoning is not only fundamental but also the starting point of the history of logical research. Logicians extend syllogistic statements into propositional statements and further into first-order logic and modal logic. Psychologists take syllogistic reasoning as the microcosm of human rationality. Thus, syllogistic reasoning is a subset or a primitive version of other complex logical forms. If supervised neural networks cannot achieve the symbolic level rigour of a subset or a primitive version of logical reasoning, they cannot achieve the rigour of logical reasoning. We do not overstate.
>
> 2. The experiments rely on controlled and synthetic inputs (circles, hypernym pairs, etc.), which may not reflect the complexity or ambiguity of real-world reasoning contexts. Extending the analysis to noisy or real-world data could provide stronger evidence for generalization.
>
> The analysis of noisy or real-world data can be carried out by introducing the fuzzy boundary of spheres. The controlled and synthetic inputs (circles, hypernym pairs) can be understood as the basic form or prototypes. If supervised neural networks cannot achieve the symbolic level rigour of logical reasoning in these basic forms, they cannot achieve the rigour of logical reasoning in more complex forms in the real-world.
>
> 3. The evaluation of GPT-5 and GPT-5-nano, while illustrative, limits reproducibility and transparency. Without open-source benchmarks, it is difficult to verify the validity of the reported results or to replicate the symbolic explanation failures.
>
> Exactly. We surveyed evaluations on open-source benchmarks from Line 152 to Line 161. There is no signal to see that open-source benchmark LLMs can reach symbolic level rigour of syllogistic reasoning. We experimented on GPT-5 and GPT-5-nano, because they are easy for public access.
>
> 4. While qualitative analysis is thorough, the statistical treatment of results (e.g., variance across trials, significance testing) is absent. ..
>
> Absolutely. We will add this point at the end of the paper as a future work item.
>
> 5. The conclusion that “scaling laws cannot reach symbolic-level reasoning” is philosophically appealing but empirically underdetermined. It might be more accurate to phrase it as a “current limitation” rather than a “fundamental impossibility,” leaving room for potential hybrid or unsupervised solutions.
>
> Yes. We will make it clear that this claim is limited to neural networks that have the two deficits: using training data, and using end-to-end mapping from premises to conclusions. Unsupervised neural networks may achieve symbolic level rigour of syllogistic reasoning -- for example, Sphere Neural Networks have achieved.
>
> 6. Can Super Euler Net or the broader analytical framework be extended to address multi-step or relational reasoning tasks such as first-order logic proofs or causal inference? How might the identified deficits manifest in these more complex settings?
>
> Yes. Syllogistic reasoning is a deduction among sets, which is a subset of first-order logic, spatiotemporal logic, and causal inference, where the entity will be a four-dimensional spatiotemporal entity. The same deficits remain in all these settings. We can transform Directed Acyclic Graphs (DAGs) into Euler diagrams (using Sphere Neural Networks).
>
> 7. Have you explored the possibility of combining your approach with formal symbolic provers...? Such hybrid systems might circumvent the “end-to-end contradiction” by separating perception from logical validation.
>
> No, we consider the reasoning capability of pure neural networks. Symbolic provers work very well for syllogistic reasoning.
>
> 8. Since syllogistic reasoning can be expressed as triplet relations (subject–predicate–object), have you considered representing reasoning structures using graph neural networks or relational transformers? These may inherently encode logical compositionality and offer a path toward symbolic consistency.
>
> The issue of logical compositionality has been addressed in Guzman et al. (2024). Here, we address the question of whether supervised neural network can reason single steps of logical deduction.
>
> 9. To what extent do you view your conclusion (...) as a theoretical impossibility versus a practical limitation? Clarifying this distinction could prevent misinterpretation and better position your work within the ongoing debate on neural-symbolic reasoning.
>
> That is a good suggestion. It is a theoretical impossibility that data-driven supervised neural networks cannot reach the symbolic level of logical reasoning. We restrict our claim to pure supervised neural networks, do not touch neural-symbolic reasoning.

---

> > ### Comment · Reviewer_Vfhs · 2025-11-28
> >
> > I appreciate the authors' rebuttal, which has addressed most of my concerns. At this stage, I intend to maintain my original score and would be interested in seeing the feedback from the other reviewers.

---

> > > ### Author Response · Authors · 2025-11-28
> > >
> > > Dear Reviewer,
> > >
> > > We are delighted that our explanation has addressed most of your concerns. We respectfully invite you to update the score in due course.

---

### Official Review · Reviewer_o5kv · 2025-10-30

**Soundness:** 1
**Presentation:** 2
**Contribution:** 2
**Rating:** 2
**Confidence:** 3

**Summary:**

To address the question of where the limit of the scaling law for logical reasoning lies, the authors identify two reasons demonstrating that all image-input neural networks and LLMs fail to achieve 100% reasoning accuracy. The first reason is that the training data cannot effectively distinguish among all 24 types of valid syllogistic reasoning. The second is that the end-to-end mapping from premises to conclusions introduces conflicting training objectives between neural components responsible for pattern recognition and those responsible for logical reasoning.

**Strengths:**

The paper employs an automated setup to evaluate the reasoning abilities of state-of-the-art LLMs, which provides a strong motivation for assessing the realistic reasoning capabilities of these models.

**Weaknesses:**

The paper aims to investigate whether data-driven machine learning systems can achieve the same level of performance as symbolic reasoning systems by increasing training data and training time. However, the authors only use a single, small benchmark consisting of images containing syllogistic reasoning information applied to specific neural networks, such as the GPT-5. As a result, the problem setting is overly broad and cannot accurately characterize by the tested benchmarks and models.

**Questions:**

1. When using Euler diagrams as examples in Figure 3, why not employ a more formal language as the benchmark to evaluate the reasoning abilities of neural networks? Furthermore, in the experiments with image inputs, the main objective appears to be evaluating the reasoning abilities of all image-input neural networks, including LLMs. How can image data adequately represent the reasoning capabilities of neural networks? Most prior research on reasoning first tackles symbolic reasoning described in formal languages before applying neural networks.
2. The authors represent four situations with statements like “some W are U” and “some W are not U,” but such symbolic statements cannot fully capture the exact logical relationships between W and U. So, the textual description statements are not good examples for indicating the logical relations in Figure 3. Why the authors use these implicit textual statements?
3. In Line 24, the authors state that their experiments illustrate limitations common to all image-input supervised networks. What types of neural network architectures were used? Can you provide a detailed list? Based on the architecture presented in Figure 4, why is this architecture considered universal for representing any neural network with image inputs?

---

> ### Author Response · Authors · 2025-11-16
>
> 1. When using Euler diagrams as examples in Figure 3, why not employ a more formal language?  Most prior research on reasoning first tackles symbolic reasoning described in formal languages before applying neural networks.
>
> Using formal language is popular and also fits with the framework of supervised deep learning.  Here, we want to find a supervised neural network that reaches the symbolic-level rigour of syllogistic reasoning. Supervised deep learning may have very good performance (almost 100%) accuracy on a testing data only when this testing data has the same distribution as the training data. If we use a more formal language in Figure 3, it will be very easy to generate testing data with rare words or rare linguistic sequences, which will cause a performance drop in this testing data. Recent evaluation of LLMs on syllogistic reasoning has shown this fact (line 152-161).
>
> 2. How can image data adequately represent the reasoning capabilities of neural networks?
>
> Before human babies can use linguistic symbols to express ideas, they can already think and reason. Animals reason very well in space and time. Recent experiments have reported that monkeys can do disjunctive syllogistic reasoning [1]. The psychological literature advocates the mental model theory of syllogistic reasoning [2], which includes the use of diagrams [3]. Therefore, diagram inputs are adequately represent the reasoning capability of neural networks.
>
> [1] Stephen Ferrigno, Yi Yun Huang, and Jessica F. Cantlon (2021). Reasoning Through the Disjunctive Syllogism in Monkeys. Psychological Science,32(2):1–9.
>
> [2] S. Khemlani and P. N. Johnson-Laird (2012). Theories of the syllogism: A meta-analysis. Psychological Bulletin, 138(3): 427–457.
>
> [3] Jill H. Larkin, Herbert A. Simon (1987), Why a Diagram is (Sometimes) Worth Ten Thousand Words, Cognitive Science, Volume 11, Issue 1, Pages 65-100.
>
> 3. The authors represent four situations with statements like “some W are U” and “some W are not U,” ... in Figure 3. Why the authors use these implicit textual statements?
>
>  “Some W are U” and “some W are not U” are two syllogistic relations between W and U. We listed the four syllogistic relations in section 2. Here, we show the many-to-many mapping between symbolic syntax relations and set-theoretic semantic relations. This causes some types of valid syllogistic reasoning cannot be well covered by the training data (experiment results are listed in Appendix C).
>
> 4. In Line 24, the authors state that their experiments illustrate limitations common to all image-input supervised networks. What types of neural network architectures were used? Can you provide a detailed list? Based on the architecture presented in Figure 4, why is this architecture considered universal for representing any neural network with image inputs?
>
> The correct interpretation of Line 24 is that any neural network architecture that has the two deficits cannot attain the symbolic-level rigour of syllogistic reasoning. The neural network architecture of Euler Net (Figure 4) is not considered universal for representing any neural network with image inputs. We can replace the Siamese components and the CNN reasoning component of Euler Net with other neural architectures, e.g., LSTM, Transformer, RNN, CNN, MLP, or any combination. As long as, the new neural network is trained by directly mapping premise inputs to the conclusion, this neural network cannot achieve the symbolic level rigor of syllogistic reasoning.
>
> 5. The authors only use a single, small benchmark consisting of images containing syllogistic reasoning information applied to specific neural networks, such as the GPT-5. As a result, the problem setting is overly broad and cannot accurately characterize by the tested benchmarks and models.
>
> We used an unpopular but elegant proof strategy. We describe our line of argument as follows.
> ```
> (1) Training data cannot cover all types of valid syllogistic reasoning.
>
> (2) Supervised neural networks will not have training data covering all types of valid syllogistic reasoning.
> ------
> Therefore, supervised neural networks will not cover all types of valid syllogistic reasoning.
> ```
> ```
> (1) A part of a premise and the whole premise can have similar vector embeddings. (Premises can be either an image or linguistic strings).
>
> (2) End-to-end mapping from premises to conclusion consists of a pipeline that encodes premises into vectors and integrates them into the conclusion vector.
> ---
> Therefore, through such an end-to-end pipeline, the whole premise and its parts can be mapped to the same conclusion vector.
> ```
> ```
> (1) In a supervised deep learning system S, premises and conclusions are embedded as vectors.
>
> (2) S learns syllogistic reasoning via end-to-end mapping from premises to conclusion
> ----
> Therefore, S may draw conclusions from partial premises.
> ```
> We will include your comment in the revision and list this line of argument in the Appendix.

---

> > ### Comment · Reviewer_o5kv · 2025-11-27
> >
> > We thank you for your responses. However, I am still wondering about the core contribution of your work and its scientific impact. Please see my detailed questions below:
> >
> > 1. What is the essence of the scaling law in your work? There is no reference or preliminary explanation regarding the scaling law in the manuscript. Could you clearly define what you mean by “scaling law” and describe the scientific algorithm, method, or technique used to implement it?
> >
> > 2. Applicability of scaling laws under LLM scenarios. If the scaling law is claimed to hold for LLMs, yet your experiments are conducted using EulerNet, is this experimental setup appropriate to demonstrate and explain how the scaling law works? If so, please clarify the rationale.

---

> > > ### Author Response · Authors · 2025-11-27
> > >
> > > ```
> > > 1. What is the essence of the scaling law in your work? There is no reference or preliminary explanation regarding the scaling law in the manuscript.  ….
> > > ```
> > > Dear Reviewer,
> > >
> > > We reference the scaling law (Kaplan et al., 2020; Bahri et al., 2024) at Line 39 and Line 194.  We will add the name of the scaling law before the first reference.
> > >
> > > (Kaplan et al., 2020) Jared Kaplan, Sam McCandlish, Tom Henighan, Tom B. Brown, Benjamin Chess, Rewon Child, Scott Gray, Alec Radford, Jeffrey Wu, and Dario Amodei. *Scaling laws for neural language models*, 2020.
> > >
> > > (Bahri et al., 2024). Yasaman Bahri, Ethan Dyer, Jared Kaplan, Jaehoon Lee, and Utkarsh Sharma. *Explaining neural scaling laws*. Proceedings of the National Academy of Sciences, 121(27), 2024.
> > >
> > > The scaling law is one of the most influential discoveries in data-driven Machine Learning (ML) systems—especially for deep learning and large models. The scaling law states that the performance of a data-driven machine learning system increases as the model size (number of parameters), the training dataset, and the training time (epochs) increase. This guides researchers in improving neural network reasoning performance by increasing the amount of training data and training time. This costs a lot of effort and resources.
> > >
> > > Considering the simple form of syllogistic reasoning, most researchers may assume that LLMs can completely solve it. The contribution of our paper is to show that we will not have supervised neural networks that achieve the rigour of syllogistic reasoning. As syllogistic reasoning is a subset of logical reasoning, they cannot achieve the rigour of logical reasoning.
> > >
> > > Our work will help practitioners improve the understanding of the scaling law and seek alternative approaches to develop highly reliable neural reasoners in high-stakes domains (Line 483 – 485) -- rigorous reasoning can be achieved without using training data and GPUs, as demonstrated by Sphere Neural Networks (Line 194 – 196).
> > >
> > > We will clarify the essence of the scaling law and related issues in our paper.
> > >
> > > ```
> > > 2. Applicability of scaling laws under LLM scenarios. If the scaling law is claimed to hold for LLMs, yet your experiments are conducted using EulerNet, is this experimental setup appropriate to demonstrate and explain how the scaling law works? If so, please clarify the rationale.
> > > ```
> > >
> > > Yes. Recent evaluations on LLMs suggest that the scaling law holds in most cases – There are exceptions. For example,  Eisape et. al. (2024) reported that PaLM 2-Small achieved better accuracy than PaLM 2-Large. This is inconsistent with what the scaling law promises (Line 154-155).
> > >
> > > (Eisape et. al. 2024) Tiwalayo Eisape, MH Tessler, Ishita Dasgupta, Fei Sha, Sjoerd van Steenkiste, and Tal Linzen. A systematic comparison of syllogistic reasoning in humans and language models. In NAACL, 2024.
> > >
> > > Our experiments with Euler-Net show that increasing the amount of training data nd training time will improve its reasoning performance. Our Super Euler-Net can automatically generate new training data in each iteration. After 19 iterations, the reasoning performance of Euler-Net improved from 56.0% to 97.8%.
> > >
> > > The scaling law is claimed to hold for all supervised neural networks, including LLMs, Euler-Net.
> > >
> > > We will make our experimental setup clearer.

---

### Meta-Review · Area_Chair_maFh · 2026-01-05

**Summary:**

The paper investigates the scaling law i.e. whether data-driven neural networks can achieve symbolic-level logical reasoning solely through increased data and computational scale. Since LLMs are the major scaled models currently, the authors study them along with image-input supervised networks and show that all these neural networks and LLMs fail to reason in a perfect manner. The paper received 4 reviews with  1 reviewer leaning positive and 3 reviewers leaning towards rejection.

The major points of contention for the reviewers were as follows:

* Weak experimental section. Majority of the reviewers were of the opinion that results on a single synthetic data set is not enough to put across the message the apaper wants to convey. Also the complexity or ambiguity of real-world reasoning context might not be well captured in these synthetic experiments.

* Other frameworks than Euler Nets can and should also be considered especially some neuro-symbolic methods which are a de-facto for complex reasoning in ML literature.

**Reviewer Concerns:**

I think the experimental section is indeed a bit weak. Symbolic reasoning is a "hot" topic of reasearch currently with a lot of benchmarks that could have been used for experimentation in the paper. This would have made the paper stronger empirically as well as shown generalization on atleast a couple of varying tasks. The authors mention in the rebuttal and I quote: "The analysis of noisy or real-world data can be carried out by introducing the fuzzy boundary of spheres" but I none of such experiments are reported in the revised version of the paper.

Also the rebuttal for considering on Euler Nets is a bit weak in my opinion. NeSy systems have shown strong results in similar benchmarks such as CLEAR, CLEARER, Bongard-HOI etc. Thus, I think the answer by the authors is a bit unsatisfactory.

**Reviewer Scores:**

Overall, I do not see reviewers changing their score considerably. Even the reviewer who mentions changing the score says that they will increase it slightly which means 2->3 or 4 at best. I concur with all reviewers here that more work is required in order for the work to be accepted.

---

### Decision · Program_Chairs · 2026-01-26

Reject